# Hormonal Regulation of Mammalian Adult Neurogenesis: A Multifaceted Mechanism

**DOI:** 10.3390/biom10081151

**Published:** 2020-08-06

**Authors:** Claudia Jorgensen, Zuoxin Wang

**Affiliations:** 1Behavioral Science Department, Utah Valley University, Orem, UT 84058, USA; 2Psychology Department and Program in Neuroscience, Florida State University, Tallahassee, FL 32306, USA; zwang@psy.fsu.edu

**Keywords:** adult neurogenesis, hormones, hippocampus, dentate gyrus

## Abstract

Adult neurogenesis—resulting in adult-generated functioning, integrated neurons—is still one of the most captivating research areas of neuroplasticity. The addition of new neurons in adulthood follows a seemingly consistent multi-step process. These neurogenic stages include proliferation, differentiation, migration, maturation/survival, and integration of new neurons into the existing neuronal network. Most studies assessing the impact of exogenous (e.g., restraint stress) or endogenous (e.g., neurotrophins) factors on adult neurogenesis have focused on proliferation, survival, and neuronal differentiation. This review will discuss the multifaceted impact of hormones on these various stages of adult neurogenesis. Specifically, we will review the evidence for hormonal facilitation (via gonadal hormones), inhibition (via glucocorticoids), and neuroprotection (via recruitment of other neurochemicals such as neurotrophin and neuromodulators) on newly adult-generated neurons in the mammalian brain.

## 1. Introduction

Adult neurogenesis, resulting in adult-generated functioning neurons, is still one of the most captivating research areas of neuroplasticity. While the first accounts were met with decades of skepticism, methodological advances—including the introduction of the synthetic thymidine analog 5-bromo-3′-deoxyuridine (BrdU) and the use of cell-type specific markers—helped to establish neurogenesis in adult rodents [1,2]. Adult-generated neurons have also been found in numerous other species, including marmosets [3,4], macaques [5], opossums (*Monodelphis domestica*) [6], and even humans [7,8]. Ultimately, adult neurogenesis was accepted as a real phenomenon and has since been observed in almost all mammals examined so far [9,10], but see [11].

Adult-generated neurons have been most commonly observed in the hippocampus and the main olfactory bulb (MOB, see Figure 1). Specifically, cells generated in the subgranular zone (SGZ) migrate to the granular cell layer (GCL) of the dentate gyrus (DG) of the hippocampus where many mature into functional granule cells [12,13]. Cells generated in the subventricular zone (SVZ) of the lateral ventricles migrate along the rostral migratory stream (rms) to mature into interneurons in the MOB [13,14,15]. Adult neurogenesis has also been documented in other brain regions such as the amygdala (AMY) [16,17,18,19], bed nucleus of the stria terminalis [18], hypothalamus (HYP) [17,19,20,21], medial preoptic area (MPOA) [18], neocortex [22,23,24], piriform cortex [16], and the striatum [18,22,25].

The addition of new neurons in adulthood—regardless of species or brain region—follows a seemingly consistent and complex multi-step process [26,27]. Briefly, these neurogenic stages include proliferation, differentiation, migration, maturation/survival, and integration into the existing neuronal network (for more detail see [9]). Proliferation refers to the birth of new cells, which then undergo glial or neuronal fate specification [28]. The birth of new cells is commonly assessed using endogenous markers (such as Ki67) or exogenous markers (such as 5-bromo-3′-deoxyuridine, BrdU, or tritiated thymidine combined with a survival time of 2–24 h) [26,29,30,31,32,33,34,35]. Cells that have undergone neuronal differentiation express morphological characteristics of neurons as well as neuronal markers (such as doublecortin, Dcx; neuron-specific class III β-tubulin; and NeuN) [36,37,38]. Following migration to a specific brain region, immature neurons develop dendrites and an axon—steps necessary to the survival of the neuron. Survival of adult-generated cells can be investigated by BrdU injection and perfusion any time after 25 h [28]. Lastly, the adult-generated neuron forms synapses to allow connections with the surrounding neuronal network [23,39]. Adult-generated neurons that fail to integrate into the existing neuronal network are eliminated by apoptosis (assessed by apoptotic markers or pyknotic morphology) [40,41,42]. Evidence suggests that cell birth and cell death in adulthood, as observed during development, are closely coordinated events [42,43,44].

Numerous exogenous (e.g., voluntary exercise and exposure to environmental enrichment) and endogenous (e.g., hormones and neurotrophins) factors influence these different neurogenic stages [13,45,46,47]. Specifically, a factor influencing cell proliferation either up- or down-regulates the birth of new cells, while a factor influencing survival promotes or prevents differentiation, maturation, and/or integration. Interestingly, the individual neurogenic stages might be influenced independently of one another. As such, cell proliferation can be up-regulated without influencing the other stages or cell survival might be down-regulated without altering other stages. Therefore, it is essential to investigate the impact of exogenous and endogenous factors on each of the neurogenic stages. While numerous reviews have discussed the impact of hormones such as gonadal steroids [48,49,50,51,52,53,54] and glucocorticoids [49,55,56,57] on adult neurogenesis, the following review will highlight the multifaceted impact of hormones on the various stages of mammalian adult neurogenesis. This is a fairly unique approach that to our knowledge has rarely been used. Specifically, we will discuss hormonal facilitation (via gonadal steroids), inhibition (via glucocorticoids), and neuroprotection (via the recruitment of the brain-derived neurotrophic factor, BDNF, and the neuromodulators serotonin, 5-HT, and oxytocin, OT) of mammalian adult neurogenesis. The impact specifically on the various neurogenic stages will be reviewed in the DG and the SVZ/MOB system, but nontraditional neurogenic brain regions will also be discussed.

## 2. Hormonal Facilitation of Adult Neurogenesis

Various factors have been shown to facilitate adult neurogenesis altering the different neurogenic stages (including cell proliferation, cell survival, and neuronal differentiation) independently from one another [13,45,46]. For example, voluntary exercise increased DG cell proliferation and survival, whereas exposure to an enriched environment only increased DG cell survival [58]. Therefore, it is essential to investigate the impact of any neurogenic factor on each stage of neurogenesis separately. Most studies on gonadal steroid regulation of adult neurogenesis have focused on androgens (e.g., testosterone and dihydrotestosterone (DHT)) and estrogens (e.g., estradiol, estrone, and estriol), thus we will discuss the evidence of these hormones acting as neurogenic factors in the adult mammalian brain—highlighting the specific stages of adult neurogenesis that can be impacted.

### 2.1. Androgens

Androgens are hormones that influence male reproductive activity; play a role in social behavior, cognition, and mood; and are potent regulators of neural plasticity [59,60,61,62,63]. Here, we will discuss the evidence that neurogenic stages might be affected by the natural fluctuations of androgens and by manipulations of the androgen system (namely castration, TX, the bilateral removal of testes) and androgen replacement.

#### 2.1.1. Natural Fluctuations of Androgens

Mammals commonly display seasonal reproduction, which is associated with variations in blood androgen levels [64,65,66,67]. The seasonally reproductive meadow vole (*Microtus pennsylvanicus*) displays a photoperiod-dependent reproductive status—with exposure to a long photoperiod resulting in larger testicular weight and higher blood androgen levels [68,69]. Consequently, male meadow voles that display high androgen levels during the breeding season (long photoperiod) and low androgen levels during the non-breeding season (short photoperiod) have been used as a model to study the impact of seasonally fluctuating androgen on adult neurogenesis. One such study found that cell proliferation in the hilus, but not GCL, was higher in wild-living meadow voles captured during the breeding compared to the non-breeding season [70]. No other neurogenic stages were investigated. A subsequent study [41] aimed to address factors (such as age, previous experience, and capture-induced stress response) that may function as potent regulators of adult neurogenesis [71,72,73,74] and cannot easily be controlled in a wild sample [70]. To assess hippocampal cell proliferation and survival, laboratory-reared meadow voles were acclimated to a long or short photoperiod to simulate the breeding or non-breeding season, respectively [41]. Higher DG cell survival, but not proliferation, was found in reproductively-active versus -inactive males. As laboratory-reared voles likely lack the same complex demands as wild-living voles, another study used endogenous adult neurogenesis markers—eliminating the need for captivity—in wild-living meadow voles to assess DG neurogenesis [69]. In this study, reproductively-active males displayed less cell proliferation and neuronal differentiation in the GCL and SGZ than reproductively-inactive voles. It is important to mention that this study was solely correlational (with many uncontrolled variables such as age or the heightened glucocorticoid levels during the breeding season). Therefore, future studies should verify the relationship of androgens on all neurogenic stages experimentally. Collectively, the findings from these studies suggest that prolonged exposure to high circulating androgen levels during the breeding season inhibit cell proliferation but enhance cell survival in meadow voles (see Table 1).

Blood androgen levels also change due to sexual experience—increasing before, during, and following sexual activity, which, in turn, can impact adult neurogenesis [92,93,94,95]. Indeed, one acute mating encounter increased DG, but not SVZ, cell proliferation in male Sprague-Dawley rats and cell survival in the accessory olfactory bulb (AOB), but not MOB, of male Wistar rats without altering neuronal differentiation [75,76,77,78]. Interestingly, one acute mating encounter in male C57BL mice did not alter cell survival in the AOB or MOB, but it increased neuronal differentiation in the glomerular cell layer of the MOB [79]—suggesting a potential species difference. Alternatively, methodological differences might explain the contrasting findings (sexually experienced rats [77] versus sexually naïve mice [79]). 

Chronic mating exposure increased DG cell proliferation and survival in male Sprague-Dawley rats [75,76] as well as DG cell survival in male CD1 mice [80] without altering neuronal differentiation. The majority of these DG-generated cells displayed a neuronal phenotype [75,76,80]. It is noteworthy that this upregulation of adult neurogenesis occurred even though the initial mating-induced testosterone peak had returned to baseline [75]—supporting a previously observed dissociation between sexual behavior and circulating testosterone levels [96]. In contrast, chronic mating exposure did not alter adult neurogenesis in the mating circuitry of male Syrian hamsters (*Mesocricetus auratus*) [15]. It is not clear at this time whether these contradictory findings between the DG and the mating circuitry reflect differences in methodology (daily versus weekly mating exposure), species (rat and mouse versus hamster), or brain region (DG versus mating circuitry).

Taken together, the above-mentioned studies suggest that fluctuating androgen levels impact adult neurogenesis in a neurogenic stage-specific manner. Higher androgen levels facilitate hippocampal adult neurogenesis, particularly cell survival, but not proliferation or neuronal differentiation. The impact of androgens on adult neurogenesis is also brain region-specific, as androgens did not alter adult neurogenesis in the SVZ or in the mating circuitry. 

#### 2.1.2. Castration (TX)

TX reduces circulating androgen levels profoundly, and TX is accompanied by the loss of mating behavior [97,98]. Only one study to our knowledge has examined the impact of short-term castration on DG adult neurogenesis [81]—showing that GCL and hilar cell proliferation was not altered following TX in Sprague–Dawley rats.

In contrast, long-term TX reduced cell proliferation in the GCL and SGZ, but not hilus, in male Sprague-Dawley rats [82]. Similarly to the DG, cell proliferation in the mating circuit of male Syrian hamsters was reduced following long-term TX [15]. Interestingly, long-term TX did not alter DG cell proliferation in male BALB/c and C57BL/6J mice [84,85]—suggesting a possible species difference. In addition, long-term TX also reduced cell survival in the GCL and SGZ in male Sprague-Dawley rats [81,82,83]. However, long-term TX did not alter cell survival in the hilus of male Sprague-Dawley rats or the mating circuit of male Syrian hamsters [15,81]—suggesting that the effect might be brain region-specific. Furthermore, neuronal differentiation seems to display a species-specific regulation. Namely, long-term TX did not alter neuronal differentiation in the hippocampus of male Sprague-Dawley rats or BALB/c mice but decreased neuronal differentiation in male C57BL/6J mice [82,83,84,85]

Together, these data suggest that long-, but not short-term, castration negatively impact adult neurogenesis, and this effect appears to be brain region-, species-, and neurogenic stage-specific (see Table 1).

#### 2.1.3. TX and Replacement with Androgen

Following TX, androgen replacement commonly occurs via two types of androgens (testosterone and testosterone propionate) or via testosterone metabolites 5α-dihydrotestosterone (DHT) and estradiol [99]. Short-term androgen replacement increased cell proliferation in the cortical and medial AMY of male meadow voles without altering the number of adult-generated cells in the central AMY, DG, or HYP or AMY neuronal differentiation [86]. Interestingly, DG cell survival depended on the time point of estrogen benzoate replacement (see Table 1) [87].

Long-term androgen replacement increased survival in the GCL, but not hilus, in male Sprague-Dawley rats without altering GCL neuronal differentiation [81,90]—suggesting a brain region-specific effect. The effect of testosterone replacement also appears to be dose-dependent. While testosterone doses of 0.5 and 1 mg (resulting in hyperphysiological levels) increased GCL survival, a low (0.25 mg, resulting in a level similar to gonad-intact males) or high (100 mg/pellet) dose failed to alter GCL proliferation and survival [81,88]. Additional support for the dose-dependent effect comes from in-vitro studies [100,101]—showing enhanced neurite outgrowth with lower testosterone concentrations and apoptosis with higher concentrations. The length of hormonal replacement also seems to matter, as only 30-day, but not 15- or 21-day, treatment increased DG cell survival in male Sprague-Dawley rats [83,89]. TX slowly leads to the complete elimination of sexual behavior and testosterone replacement only leads to the full recovery of mating after 8 weeks [97,102]—providing support for the long-term impact of hormonal replacement.

Long-term estradiol treatment did not promote DG neurogenesis in male castrated rats [81,90]. Long-term DHT increased GCL, but not hilus, cell survival in male Sprague-Dawley rats without altering neuronal differentiation [81]. Interestingly, this increase was not observed in middle-aged (11–12 month-old) Sprague-Dawley rats [91]. Finally, DHT treatment in rats pre-treated with an androgen receptor antagonist failed to show the DHT-induced hippocampal adult neurogenesis [103]. Therefore, the negative impact of TX on adult hippocampal neurogenesis can be reversed by long-term androgen replacement via activation of androgen receptors (see Table 1).

### 2.2. Estrogens

Estrogens are hormones that influence motivated behaviors and various cognitive functions [60,104,105]. They are also potent regulators of neural plasticity, play a role in neuronal excitability, and are involved in synaptogenesis via dendritic spine synapse formation [104,106,107]. Here, we will discuss the evidence that neurogenic stages might be affected by the natural fluctuations of estrogens and by manipulations of the estrogen system (namely ovariectomy, OVX, the bilateral removal of ovaries) and estrogen replacement. 

#### 2.2.1. Natural Cyclic Fluctuations of Estrogen

Estrogen levels fluctuate significantly across the female estrous cycle. During diestrus, the 17β-estrogen level increases gradually, rises to its maximum level in proestrus, and subsequently decreases and reaches its lowest level near the end of estrus [108]. Using the female Sprague-Dawley rat, a spontaneous ovulator that displays a continuous cycling of reproductive hormones, researchers found that rats injected with BrdU during proestrus (highest estrogen levels) displayed higher DG, but not SVZ, cell proliferation than females injected during other phases of the estrous cycle [109]. Interestingly, such alterations in DG cell proliferation across the estrous cycle were not observed in female C57BL/6 or BALB/c mice [85,110]—suggesting a possible species difference between mice and rats (see Table 2). Cyclic estrogen levels also affect DG cell survival (assessed 4, 7, 14, and 21 days following BrdU injection)—female Sprague-Dawley rats showed higher DG cell survival during proestrus than estrus [109,111]. This difference remained until 21 days, at which point the difference in DG cell survival across proestrus and estrus was no longer present. It was noted that the majority of adult-generated cells were neurons and neuronal differentiation was not altered [109,111].

Unlike female rats or mice, female meadow and prairie voles are induced ovulators, in which the exposure to a male or male pheromones elicits behavioral estrous [128,129,130]. It is of interest to note that meadow and prairie voles (*Microtus ochrogaster*) display remarkable differences in social behaviors and life strategy. Meadow voles are promiscuous [131], whereas prairie voles are socially monogamous and form lasting pair-bonds [132]. In the wild, female voles have low blood estrogen levels during the non-breeding season, but once primed, their blood estrogen levels remain elevated throughout the breeding season [133]. Researchers captured wild-living female meadow voles across breeding seasons and contrary to findings in mice and rats found that reproductively-active females displayed lower GCL and hilus cell proliferation than reproductively-inactive females [70]. A possible explanation for the discrepancy in findings between mice/rats and meadow voles might be the differences in the ovulation onset—spontaneous versus induced ovulation. Alternatively, captive housing of wild-living meadow voles might have introduced confounding variables Using endogenous markers to eliminate the need for captive housing (a potential confound), researchers observed that reproductively-active females showed lower cell proliferation and neuronal differentiation in the GCL and SGZ than reproductively-inactive females [69]. Other confounding variables might include age, previous experience, and pregnancy status—as all wild-captured female meadow voles during the breeding season were pregnant. Not surprisingly, pregnancy (which is characterized by dramatic fluctuations in steroid hormones [134]) and age (which is associated with changes in circulating 17β-estradiol levels [135]) have previously been identified as potent modulators of adult neurogenesis [71,72,73,136,137]. To address these confounding variables, researchers used laboratory-reared meadow voles and exposed them to a male (to induce behavioral estrous) or female conspecific (control) [112]. Male-exposed females were considered reproductively-active, conversely female-exposed females were considered reproductively-inactive. Reproductively-active females displayed lower GCL cell proliferation and survival than reproductively-inactive females. When the rates of adult-generated cells were compared between the proliferation and survival time points, the data indicate that high estrogen levels might have enhanced cell survival.

Using female prairie voles, researchers found that primed (via short-term male pheromone exposure) females displayed an increase in cell proliferation in the SVZ and along the rms [113]. Interestingly, in another study short-term male exposure did not alter cell proliferation in the SVZ, AMY, caudate putamen, cingulate cortex, DG, or HYP [17]. These contradictory findings between the two studies [17,113] might be due to methodological differences including the type of exposure and the type of control group used. Specifically, one study [113] used a fine wire mesh that resulted in animals being able to see, smell, and have limited physical contact with, while preventing mating. The animals in the other study [17] were housed in the same cage allowing unrestricted social interaction including mating behavior. Long-term male exposure (allowing unrestricted social interaction) increased cell survival in the AMY and HYP but not caudate putamen, cingulate cortex, DG, or MOB of female prairie voles [17]. In the DG and SVZ/MOB, the majority of adult-generated cells expressed a neuronal phenotype. There were no group differences in neuronal differentiation.

Taking together, these studies suggest that fluctuating estrogen levels impact adult neurogenesis in a species-specific manner (see Table 2). In some species (e.g., rat and prairie vole) high estrogen levels are associated with a facilitation of adult neurogenesis. In other species (e.g., meadow vole and mouse) high estrogen levels are linked to a reduction or no alteration of cell proliferation and survival.

#### 2.2.2. Ovariectomy (OVX)

OVX reduces circulating estrogen levels [107] as well as the number of estrogen receptors (ER; beta, but not alpha) [138]. Short-term OVX caused a drastic reduction in DG cell proliferation in female Sprague-Dawley and Wistar rats [109,114], whereas long-term OVX did not alter DG cell proliferation in female Sprague-Dawley, Wistar, and Long-Evans rats [111,114,115]. Long-term OVX did not alter DG cell proliferation or neuronal differentiation in C57BL/6 mice [110]. Interestingly, in a different mouse strain, BALB/c, long-term OVX reduced cell proliferation and neuronal differentiation [85].

To summarize, short-term depletion of estrogen negatively impacts DG adult neurogenesis, while long-term depletion might have a species-specific impact (see Table 2). This time-dependent manner of OVX on adult neurogenesis mirrors the results of OVX on hippocampal dendritic spine density [107]. Specifically, spine density decreases gradually for the 6 days following OVX. No further decrease is observed up to 40 days following OVX.

#### 2.2.3. OVX and Replacement with Estrogen

Following OVX, estrogen replacement might occur via three main forms of estrogen, namely estrone (E1), estradiol (E2, which includes the optical isomers 17β-estradiol and 17α-estradiol), and estriol (E3). The vast majority of studies examining neuroplasticity have used estradiol and its analog estradiol benzoate, as it is the most prevalent and potent form of estrogen [108,125].

Short-term estrogen replacement increased DG cell proliferation in female Sprague-Dawley rats and meadow voles—thereby reversing the OVX-induced reduction in cell proliferation [109,111,116,117,118,119,120,121]. Similarly, short-term estrogen replacement increased cell proliferation in the prairie vole SVZ, but not the rms or MOB [113]—suggesting a potential brain region-specific effect. Interestingly, estradiol replacement in female C57BL6/J mice led to a decrease in SVZ cell proliferation [122]—suggesting a potential species-specific effect. Alternatively, this difference might be due to species differences in baseline hormonal levels. Voles, which are induced ovulators, exhibit consistently high levels of circulating estrogen during the breeding season (20–30 days), while rats and mice exhibit high estrogen fluctuations across a 4-day estrous cycle [49]. Therefore, estrogen treatment does not cause highly unnatural hormonal levels in voles.

The effects of estrogen replacement also seem dose-specific as a 10 μg dose (that results in circulating estrogen levels in the proestrus range [139,140]) increased whereas other doses (such as 0.3, 1, or 50 μg) did not alter DG cell proliferation and a high dose (100 μg/100 g body weight) reduced cell proliferation in the AOB [111,120,123]. It is noteworthy to understand that dose and type of estrogen are related factors—as one dose might be effective for one type of estrogen while ineffective for another type of estrogen [120]. Underlying pharmacokinetic and pharmacological differences between the different types of estrogen might be causing these differences [108]. For example, the administration of estrogen esters yields different peak estradiol levels—higher levels following estradiol valerate and benzoate treatment than estradiol cypionate treatment [141]. In addition, the impact of estrogen replacement also appears time-specific. On one hand, the estrogen-induced reversal of OVX-induced reduction in cell proliferation is transient—short-term (30 min or 4 h) estrogen replacement increases whereas long-term (48 h) estrogen replacement decreases hippocampal cell proliferation [116,117]. On the other hand, latency of estrogen replacement following OVX seems to matter. Following brief latency (1 week) estrogen replacement increased, whereas long latency (28 or more days) does not alter hippocampal cell proliferation [111,124].

Long-term estrogen treatment did not alter hippocampal cell proliferation in female and male rats [90,111,125,126]—regardless of estrogen-type, dose, or sex. Interestingly, long-term estrogen replacement altered hippocampal cell survival in a sex-specific manner. Namely, hippocampal cell survival was not altered in males but reduced in females [90,126]. Interestingly, the impact on cell survival might be estrogen type-specific. Long-term treatment with estradiol benzoate and estrone reduced hippocampal cell survival, whereas 17β-estradiol treatment increased hippocampal cell survival in female rats [90,125,126]. It is worth mentioning that another study (using 17β-estradiol) found an increase in cell survival in the VMH, arcuate nucleus of the hypothalamus, and the dorsal medial hypothalamus of mice following chronic estrogen treatment [127]. At the moment, it is not known whether this finding indicates a brain region-, species-, or estrogen type-specific impact.

To summarize, estrogen impacts cell proliferation in a dose-, estrogen-type-, time-, and brain region-specific manner (see Table 2). There is also a species-specific effect—for example data showing that estrogen replacement increased AMY cell proliferation in the promiscuous meadow voles but did not alter AMY cell proliferation in the pair-bonding prairie vole [124]. However, it should be noted that the various methodological approaches across studies make it difficult to derive patterns or conclusions with certainty. It cannot be ruled out that methodological differences (e.g., type of estrogen, subjects’ ages, and estrogen dosages) might also have influenced the alterations in adult neurogenesis.

## 3. Hormonal Inhibition of Adult Neurogenesis

Exposure to stressors, such as predation, is a ubiquitous part of the animal kingdom and commonly triggers a stress response. In turn, the stress response causes the activation of the hypothalamic-pituitary-adrenal (HPA) axis, leading to increased glucocorticoids release [89]. These steroid hormones (corticosterone in rodents and cortisol in humans) seem to play an important role in neuroplasticity—especially in limbic brain regions. In the DG, high glucocorticoid levels suppress long-term potentiation [142], cause dendritic atrophy [143,144,145,146], and can result in neuronal loss [142,144,145,147]. High glucocorticoid levels alter spine density in the AMY and cause cell loss in the prefrontal cortex [148,149,150,151,152]. Furthermore, steroid hormones also influence adult mammalian neurogenesis [57,153]. Glucocorticoid administration reduces DG cell proliferation and survival in male and female rats [154,155,156,157,158] as well as DG cell proliferation in male mice [159]. Similarly, the administration of a glucocorticoid receptor agonist, reduces DG cell proliferation [160]. On the contrary, adrenalectomy, which results in the removal of circulating glucocorticoids, leads to an increase in DG adult neurogenesis [161,162,163] and eliminates the stress-induced suppression of DG cell proliferation [164]. Other means of HPA axis inhibition reverse the stress-induced suppression of DG adult neurogenesis in male mice and rats [165,166,167]. The following section will discuss the impact of stressors that are associated with glucocorticoid release on cell proliferation, cell survival, and neuronal differentiation.

### 3.1. Hormonal Inhibition of Cell Proliferation

Exposure to various acute laboratory-specific as well as ethologically-relevant stressors reduced DG cell proliferation in numerous species, without altering SVZ cell proliferation [4,164,168,169,170,171,172,173,174,175,176,177]—suggesting a potential brain region-specific regulation (see Table 3 for detail).

Such gonadal inhibition of DG cell proliferation might be time-specific. Restraint stress reduced DG cell proliferation at 6 h, but not at time points immediately, 2 h, or 3 h after conclusion of the exposure [168,178,186]. Interestingly, the timeline for inescapable shock differed substantially—with a reduction of DG cell proliferation 7 days, but not 1 h, 1 day, or 2 days after conclusion of the exposure [173,197]. Furthermore, the gonadal inhibition appears transient—as cell proliferation returns to baseline levels (1 day following restraint and 14 days following inescapable shock) [168,173]. Evidence also supports the notion that the stressor-induced impact on DG cell proliferation might also be regulated in a sex-specific (with males potentially showing higher sensitivity in rats [171,176] but see [181]), species-specific [180], and age-dependent manner [181] (see Table 3 for more detail).

It is noteworthy that the impact of stress might be related to the intensity and/or length of the stressor. Even though the exposure to a 20-min stressor (e.g., foot-shock, predator odor, social defeat) results in a robust increase in corticosterone levels, this length of stressor does not alter DG cell proliferation in male Wistar or Sprague–Dawley rats [198,199,200]. Similarly, one acute 40-min social defeat exposure (comprised of 5 min of instigation, 5 min of defeat, and 30 min of threat) as well as three 40-min social defeat exposures do not alter DG cell proliferation in male CFW mice [183]. Using a short acute stressor (5-min of forced swimming) revealed differential impacts on DG cell proliferation dependent on the type of coping style (‘reactive’ versus ‘proactive’). Specifically, male wild house mice with a long attack latency (reactive coping style) showed a reduction in DG cell proliferation in comparison to male wild mice with a short attack latency (proactive coping style) [179]. Indeed, previous research has shown that predictability and controllability can lessen the negative consequences of stress on the brain [201,202,203] and might protect against stress-induced inhibition of adult neurogenesis [171,174].

Subchronic laboratory-specific as well as ethologically-relevant stressors resulted in a reduction of DG cell proliferation in various species [182,183,184,185] (see Table 3 for details). This stress-induced reduction in DG cell proliferation is long-lasting—as 21 days following social defeat DG cell proliferation was still reduced [185]. Interestingly, subchronic psychosocial stress did not alter AMY cell proliferation in mice—possibly suggesting a brain region-specific difference [184]. Alternatively, methodological differences might explain the contradictory findings, as the length of direct exposure to the dominant animal might create a more or less intense social defeat encounter.

Chronic stress exposure (regardless of type of stressor or length) leads consistently to a reduction in DG cell proliferation in various species [165,170,178,186,187,188,189,190]. Interestingly, 21-day exposure to daily chronic mild stress did not alter DG cell proliferation in male Sprague–Dawley rats [82,89]. It is possible to speculate that such contradictory results can be explained by methodological differences—namely rats were exposed to behavioral tests prior to cell proliferation assessment. In addition, DG cell proliferation was lowered in response to varying lengths and a variety of ethologically-relevant stressors in several mammalian species [191,192,193,194,195,196].

The impact of chronic stress might be time-dependent. Specifically, 21 days of daily foot-shock experience reduced cell proliferation 2 h, but not 24 h, after the last foot-shock [198]. Furthermore, the impact of chronic stressors might also be region-specific [82,165,195]. Interestingly, social defeat stress and subsequent isolation housing in long-tailed hamsters reduced cell proliferation in the AMY and VMH, without altering DG cell proliferation [204]. At this point, it is not clear whether the lack of an effect in the hippocampus reflects a species or a methodological difference.

### 3.2. Hormonal Inhibition of Cell Survival

Although it has not been studied in detail yet, there is evidence that acute stress exposure negatively impacts cell survival (see Table 4). To our knowledge, only one study to date assessed the impact of an acute laboratory-specific stressor on cell survival and observed a reduction in DG cell survival [174]. Similarly, studies that investigated the impact of ethologically-relevant stressors observed the reduction in DG cell survival [164,200]. It is noteworthy that stress exposure did not impact immediate survival (2-day old adult-generated cells), but had negative impacts on both short-term (7-day old adult-generated cells) and long-term (28-day old adult-generated cells) survival in male Sprague–Dawley rats [200]. The stressor intensity might play a role in the longevity of the stressor-induced reduction of DG cell survival, as predator odor exposure reduced short-term (7-day), but not long-term (21-day) survival in male Sprague–Dawley rats [164].

Subchronic and chronic stress exposure leads consistently to a reduction in DG cell survival in various species [80,82,158,165,178,191,192,195,205]—regardless of laboratory-specific or ethologically-relevant stressor (see Table 4). The majority of adult-generated DG cells display a neuronal phenotype [80,83,165,178,192,206]—suggesting that exposure to a chronic stressor reduces adult neurogenesis. Research has shown that the stress-induced reduction of cell survival might be brain region-specific [80,82,195].

It has also been shown that the type of stressor might impact cell survival in a sex-specific manner. Chronic social isolation resulted in lower levels of DG cell survival in intact female prairie voles (42 days of social isolation [195]) but did not alter DG cell survival in male Sprague-Dawley (12 or 34 days of social isolation [83,206]). Indeed, sex differences in the influence of stress on neural plasticity has previously been noted. For example, in response to chronic restraint stress males display atrophy of apical CA3 dendrites, whereas females display atrophy in basal CA3 dendrites [207]. Stress exposure also alters HPA axis functioning in a sex-specific manner [187,208]. However, at the moment it cannot be ruled out that species differences (prairie vole versus rat) or length of social isolation underlie these differences.

Interestingly, if BrdU is used to label cells prior to a subchronic (e.g., social defeat) or chronic stressor (e.g., chronic daily foot-shock exposure, chronic twice daily unpredictable stress, or daily chronic restraint stress), the rate of survival of adult-generated DG cells is not altered in male Wistar rats [170,178,185,198].

### 3.3. Hormonal Impact on Neuronal Differentiation

Stress exposure has resulted in mixed findings for its impact on neuronal differentiation. Exposure to acute stressors (such as 20-min psychosocial stress or 30 trials of uncontrollable foot-shock) as well as exposure to various chronic stressors (including 21 days of chronic restraint stress; 21 days of chronic unpredictable stress; 10 or 32 days of social isolation; and 10, 18, or 35 days of daily chronic social defeat) did not alter neuronal differentiation in male C57BL/6 mice, Sprague-Dawley rats, or Wistar rats [83,170,172,178,191,192,200,205,206]. Interestingly, other studies have found that stressors decrease neuronal differentiation. Namely, exposure to an acute stressor (30-min foot shock and 30-min restraint)—a procedure that causes long-lasting and robust increase in serum corticosterone—decreased the percentage of adult-generated DG neurons (assessed by BrdU/Dcx double-labeling) in male Balb/C mice [168,169]. Furthermore, the exposure to 80 sets of tail shock reduced DG adult neurogenesis (assessed by BrdU/NeuN double-labeling) in male Sprague-Dawley rats [174]. Similarly, exposure to a chronic stressor (21 days of daily restraint stress or daily foot-shock exposure) reduced DG neuronal differentiation (assessed by BrdU/Dcx or BrdU/NeuN double-labeling) in male CD1 mice and male Wistar rats [80,198]. Furthermore, 42 days of social isolation reduced the rate of neuronal differentiation (assessed by BrdU/NeuN double-labeling) in the DG and AMY of female prairie voles [195]. At this time, it is not clear whether these contradictory findings might possibly suggest a sex difference, strain difference, difference in stressor, or differences in methodology (e.g., BrdU/NeuN double-label vs. Dcx-labeling).

In sum, the stress-induced release of glucocorticoids is one of the most profound environmental suppressors of adult neurogenesis. Indeed, laboratory-specific as well as ethologically-relevant stressors inhibit multiple neurogenic stages in various mammalian species (including mouse, rat, tree shrew, marmoset, and macaque). Furthermore, acute, subchronic, as well as chronic stress exposure results in a potent suppressive effect on adult neurogenesis—suggesting that the stress duration may have lesser role in affecting adult neurogenesis.

## 4. Hormonal Neuroprotective Effects

In addition to facilitating adult neurogenesis, hormones might also have a neuroprotective effect on adult-generated neurons. Here, we will discuss the hormonal neuroprotection by recruiting other neurochemicals as well as the evidence of hormones ameliorating the stress-induced reduction of adult neurogenesis.

### 4.1. Hormonal Neuroprotection via Recruitment of Other Neurochemicals

Adult neurogenesis—just one aspect of the highly complex process of neuroplasticity—is not solely regulated by hormones [47]. Indeed, gonadal hormones might have neuroprotective properties in part by interacting with neurotrophic factors as well as neuromodulators [209]. Here, we will discuss the interaction of estrogen and brain derived neurotrophic factor (BDNF)—a neurotrophin that regulates adult neurogenesis. Furthermore, we will review the interaction between estrogen and serotonin (5-HT) and oxytocin (OT), as 5-HT and OT have been shown to play a role in adult neurogenesis. While very little research has been conducted on the interaction between other gonadal hormones (such as testosterone) and BDNF, 5-HT, or OT, we have included these studies into our discussion.

#### 4.1.1. Hormonal Neuroprotection via Recruitment of the Neurotrophin BDNF

BDNF, which is widely expressed throughout the mammalian brain [210], plays an important role in the development [211,212], survival [213], maintenance [212], and plasticity [214] of neurons. In particular, BDNF has been implicated in various aspects of neuronal plasticity including long-term potentiation and neuronal excitability [215], synaptogenesis and spine formation [216], and dendritic growth [217,218,219,220,221], as well as adult neurogenesis [222,223]. Chronic BDNF infusion or BDNF viral overexpression in the lateral ventricles upregulate the survival of adult-generated cells—leading to more new neurons in the MOB, striatum, HYP, and thalamus [224,225]. Similarly, chronic BDNF infusion into the hippocampus increases of survival of adult-generated neurons [226]. Furthermore, BDNF-expressing cells have been reported in various neurogenic brain regions including the amygdala, hippocampus, and HYP [227,228,229,230].

BDNF interacts with estrogen [231,232,233]. Specifically, researchers found that estrogen receptors colocalize to neurons expressing BDNF and its receptor trkB in the basal forebrain [234]. Such colocalization was also observed in the cerebral cortex, HYP, and hippocampus [235]. Furthermore, researchers found an estrogen-sensitive response element on the *BDNF* gene [236]—which allows estrogen to have a direct genomic impact on BDNF expression. Consistent with this notion, it has been found that BDNF mRNA levels and BDNF immunoreactivity in the hippocampus vary throughout the estrous cycle in the rat [237,238,239]. While estrogen administration increases the expression of BDNF and its receptor in the cortex, MOB, and hippocampus [236,240,241]; OVX leads to a noticeable reduction of BDNF mRNA levels in the hippocampus, AMY, cerebral cortex, and MOB [236,240,241,242,243]. Interestingly, this OVX-induced reduction in BDNF mRNA levels can be reversed by estrogen replacement following OVX [236,237,240,241,242,243,244]. Estrogen-treated animals also show more retrograde transport of BDNF in forebrain circuits—a mechanism by which BDNF exerts its neuroprotective role [245]. While the specific underlying mechanism is not fully understood, these results taken together convincingly suggest an interaction between estrogen and BDNF. It can further be speculated that this interaction might be involved in mediating adult neurogenesis.

#### 4.1.2. Hormonal Neuroprotection via Recruitment of 5-HT

5-HT regulates diverse brain functions such as autonomic nervous system reactivity, sleep cycles, and appetite [246,247]. Furthermore, 5-HT has a role in regulating various emotional behaviors such as anxiety, aggression, and affiliative behaviors [248]. Serotonergic projections originating from the brain stem raphe nuclei innervate nearly every part of the forebrain including the HYP, AMY, prefrontal cortex, and hippocampus where the 5-HT effects are mediated via 15 different 5-HT receptors [249].

5-HT seems to play a role in the regulation of adult neurogenesis [250,251,252]. Specifically, the depletion of serotonin (by ablating 5-HT neurons using a 5-HT neurotoxin) reduced DG cell proliferation in male and female Wistar rats, but not male Lister hooded rats [114,253,254,255]. Lesion-induced reduction in cell proliferation in rats was reversed by using fetal 5-HT grafts [254]. Acute fluoxetine (a 5-HT agonist by selectively inhibiting 5-HT reuptake) treatment did not alter DG cell proliferation in male Sprague-Dawley rats [256]. Interestingly, the direct manipulation of 5-HT receptor activity via receptor agonists or antagonists resulted in oppositional effects on mediating DG cell proliferation. The acute activation of the 5-HT_1A_ receptor increased DG and SVZ cell proliferation in male Wistar rats and female C57Bl/6 mice [257,258], whereas the acute blockade resulted in a reduction of DG cell proliferation in male Sprague-Dawley rats [259]. Interestingly, the acute blockade of 5-HT_2_ receptors mirrored the effects of 5-HT_1A_ receptor activation [258]. While acutely activating the 5-HT_1B_ receptor did not alter DG cell proliferation in male Wistar rats [257], it reduced SVZ cell proliferation in female C57Bl/6 mice [258]—suggesting a potential brain region- or species-specific effect. Further support for a species-specific effect of the regulation of 5-HT receptor activity comes from the finding that a 5-HT_2C_ agonist did not alter DG cell proliferation in male Wistar rats [257] but reduced DG cell proliferation in female C57Bl/6 mice [258]. To our knowledge, there is only one study to date that assessed the impact of acute 5-HT system manipulation on cell survival. The researchers noted that an acute treatment with a partial 5-HT_1A_ agonist increased the survival of adult-generated MOB and DG neurons in opossums [6]. Similarly, there is only one study we are aware of that investigated the impact on neuronal differentiation. The researchers found that the activation of 5-HT_1A_R increased the number of Dcx-labeled cells in the hippocampus [258].

Subchronic (5 days) depletion of serotonin (by inhibiting 5-HT synthesis) reduced cell proliferation in the DG and SVZ of male Wistar rats [260]. Subchronic (7 days) activation of 5-HT_1A_ receptor had no effect on DG cell proliferation, whereas treatment of a 5-HT_1A_ receptor antagonist reduced the number of BrdU-labeled cells in the DG [258]. Furthermore, subchronic treatment with 5-HT_1A_R agonist did not alter the number of Dcx-labeled cells in the DG of female C57Bl/6 mice [258].

Chronic treatment (11, 14, 21, 28, 42, or 63 days) with fluoxetine increased DG adult neurogenesis in male Sprague-Dawley rats, Brown-Norway rats, Lister hooded rats, Wistar rats, 129/SV, and C57BL/6 mice [110,256,261,262,263,264,265]. It should be noted that 25 days of fluoxetine treatment in male aged Sprague-Dawley rats (12 months old) and male aged C57BL/6 mice (6 or 12 month old) did not alter DG cell proliferation [266,267]. Interestingly, researchers observed that chronic fluoxetine treatment for 42 and 63, but not 21, days reduced SVZ cell proliferation in male C57BL/6J mice [268]—suggesting that 5-HT might mediate adult neurogenesis differently across brain regions (SVZ versus DG) and potentially differently across species (rat versus mouse) [269]. The administration of a 5-HT_1A_ agonist delivered via an osmotic pump, but not daily injections, for 14 days increased DG cell proliferation in male Lister hooded rats [255,264].

Chronic treatment (28 days) with fluoxetine increased DG cell survival in adult (3 month old), but not aged (6 or 12 month old) male C57BL/6 mice [267]. The majority of these adult-generated cells expressed a neuronal phenotype—suggesting that chronic fluoxetine treatment increased DG adult neurogenesis. Similarly, chronic treatment (28 days) with fluoxetine increased the number of immature DG neurons in male adult, but not aged, Wistar rats [263,266]; and chronic treatment (14 days) with a partial 5-HT_1A_ agonist increased the number of immature neurons in the hippocampus of male Sprague-Dawley rats [270]. Further, a 14-day treatment with 5-HT_1A_ agonist delivered via an osmotic pump, but not via daily injections, increased hippocampal cell survival and showed a higher number of neuronal hippocampal cells than saline-treated animals in male Lister hooded rats [255,264].

Various studies suggest that ovarian steroids, such as estrogen, interact with the 5-HT system [104]. One example of such an interaction is the localization of estrogen receptors to 5-HT neurons in various species including guinea pigs, macaques, mice, and rats [271,272,273,274,275]. Furthermore, there is evidence that estrogen affects the function of the 5-HT system [276]. Acute estrogen treatment (32-h duration) increased levels of 5-HT_2A_ mRNA in the dorsal raphe of male castrated rats [277]. Chronic estrogen treatment also decreased the expression of 5HT_2c_ receptors mRNA in the HYP of spayed female pigtail macaque (*Macaca nemestrina*) without altering the expression of 5-HT_1A_ or 5-HT_2A_ mRNA [278]. Interestingly, chronic estrogen treatment in female OVX rats reduced 5HT_1A_ mRNA in the hippocampus [279]. These contradictory results between these two studies [278,279] might suggest a possible brain-region specific effect or a species difference. Another example of the interaction of estrogen and 5-HT involved the main process to terminate 5-HT neurotransmission, the 5-HT reuptake transporter (SERT). Estrogen treatment in the female rhesus monkey raphe nuclei reduced the expression of 5-HT transporter (SERT) mRNA [280]—suggesting that estrogen can alter 5-HT neurotransmission.

A potential underlying mechanism for the enhancing effect of 5HT on adult neurogenesis is the role of 5-HT in the regulation of BDNF mRNA. For example, using a selective 5-HT reuptake inhibitor causes an increase of BDNF mRNA in the hippocampus [281]. Furthermore, the stress-induced reduction of BDNF mRNA in the hippocampus was prevented by the pretreatment with a 5-HT antagonist [282].

While more research needs to directly assess the estrogen stimulation of hippocampal cell proliferation via 5-HT, researchers found that the administration of a precursor to 5-HT can reverse the reduction in hippocampal cell proliferation following OVX, whereas estrogen treatment was unable to reverse the OVX-induced reduction in cell proliferation in rats treated with a 5-HT antagonist [114].

#### 4.1.3. Hormonal Neuroprotection via Recruitment of OT

OT is released during sexual activity and it plays an essential role in facilitating sexual and affiliative behavior, including the development of a pair bond [283,284,285]. Furthermore, OT contributes significantly to the initiation of maternal behavior, regulates the selective bond between mother and offspring, and might play an important role in paternal behavior [286,287,288]. OT is primarily produced in the HYP, which projects to the pituitary gland as well as to various regions within the brain [283]. Some evidence has accumulated that OT might be a factor influencing adult neurogenesis.

Indeed, an acute peripheral or central (into the hippocampus) OT administration upregulated DG cell proliferation in male rats without altering cell proliferation in the SVZ [289]. This effect was dose-dependent as 1 mg/kg dose led to an increase in cell proliferation, whereas 10 mg/kg caused no change. Subchronic OT treatment increased DG adult neurogenesis [289]. An acute OT administration did not alter the survival of adult-generated DG cells as assessed 1 or 3 weeks following OT administration [289].

Subchronic peripheral OT administration increased the survival of adult-generated DG cells in male rats [289]. The majority of these cells expressed a neuronal phenotype. Subchronic peripheral OT administration did not alter neuronal differentiation in the DG of male rats [289].

Various studies suggest that ovarian steroids, such as estrogen, interact with the OT system. Specifically, researchers found that estrogen receptors (specifically ER beta) colocalize to OT-producing neurons in the HYP [290,291]. Natural fluctuations of the gonadal hormones influence the OT system. Specifically, prior to parturition, which, leads to an increase in estrogen [134], OTR expression is increased [292]. Additionally, estrogen and testosterone treatment increase OTR binding and OT mRNA levels in the brain, whereas these levels are reduced following castration [293,294,295]. Future studies should investigate the underlying mechanism for this OT mediation of adult neurogenesis by investigating whether OT receptors are expressed by proliferating precursor cells as well as what mechanism estrogen has.

### 4.2. Hormonal Amelioration of Stressor-Induced Reduction of Adult Neurogenesis

Exposure to stressors and the associated upregulation of glucocorticoids have been shown to downregulate neurogenesis. There is evidence to suggest that gonadal hormones might mediate the impact of the stress response on adult neurogenesis.

Stress exposure or corticosterone administration inhibit adult neurogenesis and alter the functioning of the BDNF, 5-HT, and OT systems [296,297,298]—chemicals involved in the neuroprotection of adult-generated neurons. Interestingly, gonadal steroids can confer resiliency to the stress-induced reduction of adult neurogenesis. For example, TX and chronic mild stress in male rats reduced DG cell proliferation and survival more than chronic mild stress alone [82]. Similarly, TX and isolation stress in male rats resulted in a lower survival rate of adult-generated DG neurons than isolation alone [83]. It should be noted that the majority of these adult-generated cells expressed a neuronal phenotype. Furthermore, environmental manipulation of gonadal steroids (i.e., mating exposure, which, in turn, alters the testosterone system) can buffer against the negative impact of stress in male rats [80]. Specifically, males exposed to both daily restraint stress and mating had more BrdU-labeled DG cells (indicating a higher level of cell survival) than rats which were only restraint. The majority of these cells expressed a neuronal phenotype, but the number of BrdU/NeuN double-labeled cells did not differ across groups—suggesting that mating activity buffers against stress-induced reduction of hippocampal adult neurogenesis.

Pharmacological manipulations of gonadal steroids can also buffer against the negative impact of stressors on adult neurogenesis. Estrogen treatment prevented the chronic stress-induced dendritic retraction in the hippocampal of female OVX Sprague-Dawley rats [299]. Estrogen treatment also attenuated hippocampal neuronal loss in chronically stressed female OVX rats (Takuma 2007) [300]. Testosterone treatment in male Sprague-Dawley rats prevented the reduction in hippocampal cell proliferation following social defeat stress [185].

## 5. Conclusions

The evidence we reviewed here strongly indicate that hormones have a multifaceted impact in regulating adult neurogenesis (Figure 2). We highlight that gonadal hormones seem to facilitate while glucocorticoids seem to inhibit adult neurogenesis. Furthermore, gonadal steroids have been shown to have a neuroprotective effect on adult-generated cells by interacting with BDNF, 5-HT, and OT. These findings are not surprising as the hippocampus and other neurogenic regions (such as the AMY and MOB) are enriched with receptors for gonadal hormones, prolactin, glucocorticoids, BDNF, 5-HT, and OT. However, the exact mechanisms—whether acting on astrocytes or directly on progenitor cells—for hormones to impact adult neurogenesis in such a diverse pattern still remain to be elucidated. Future studies should systematically investigate the functional implications of this multifaceted regulation of hormones on motivated behaviors. Such investigations might further elucidate the observed differences across species, brain-regions, and age of subjects.

## Figures and Tables

**Figure 1 biomolecules-10-01151-f001:**
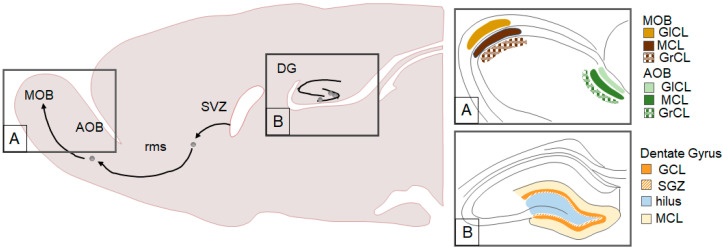
Traditional neurogenic brain regions including the subventricular zone (SVZ) system and the dentate gyrus (DG) of the hippocampus. A. Cells generated in the SVZ migrate along the rostral migratory stream (rms), pass the accessory olfactory bulb (AOB), and reach the main olfactory bulb (MOB). The MOB and AOB have the following three layers: glomerular cell layer (GlCL), molecular cell layer (MCL), and the granular cell layer (GrCL). B. The DG has various components including the granular cell layer (GCL), subgranular zone (SGZ), hilus, and molecular cell layer (MCL).

**Figure 2 biomolecules-10-01151-f002:**
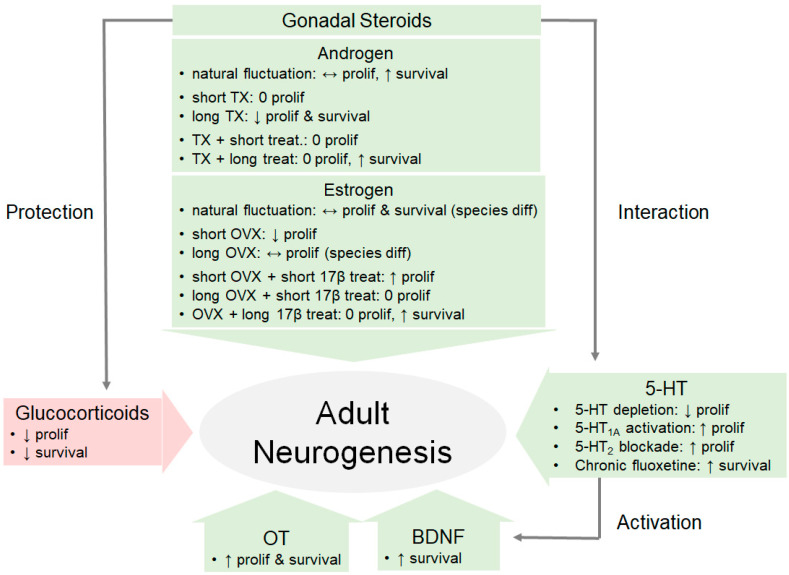
Hormones have a multifaceted impact on adult neurogenesis. This model diagram illustrates some of the effects of various hormones on hippocampal adult neurogenesis, including the facilitation by gonadal hormones, inhibition by glucocorticoids, and protection via the recruitment of other neurochemicals. BDNF, brain-derived neurotrophic factor; diff, difference; long, long-term; OVX, ovariectomy; OT, oxytocin; prolif, proliferation; 17β, 17β estradiol; short, short-term; treat, treatment; TX, castration; ↑: increase; ↓: decrease; ↔: mixed findings; 0: no change.

**Table 1 biomolecules-10-01151-t001:** The effects of androgens on the different stages of adult neurogenesis in several mammalian species.

		Species	Cell proliferation	Cell Survival	Neuronal Differentiation	References
**NATURAL INCREASE**						
Breeding season		wild-living ♂ MV	↔ GCL+SGZ	--	↓ GCL+SGZ	[69,70]
		lab-reared ♂ MV	0 GCL+SGZ	↑ GCL+SGZ	--	[41]
Sexual experience	Acute (One encounter)	♂ rat (young & middle-aged)	↑ DG 0 SVZ	↑ AOB 0 MOB	0 AOB	[75,76,77,78]
		♂ mouse	--	0 AOB, MOB	0 AOB ↑ MOB	[79]
	Chronic (Daily or weekly encounter 14 days)	♂ rat (young & middle-aged)	↑ DG	↑ DG	0 DG	[75,76]
		♂ mouse	--	↑ DG	0 DG	[80]
		♂ hamster	0 MeP, MPOA	0 MeP, MPOA	--	[15]
**DECREASE VIA TX**						
Short-term TX (7 days)		♂ rat	0 GCL	--	--	[81]
Long-term TX (>14 days)		♂ rat	↓ GCL+SGZ	↓ GCL, GCL+SGZ	0 GCL+SGZ	[81,82,83]
		♂ mouse	0 DG	--	↔ GCL	[84,85]
		♂ hamster	↓ MeP 0 MPOA	0 AMY, MPOA	--	[15]
**TX AND ANDROGEN REPLACEMENT**						
Short-term replacement (≤5 days)	Testosterone proprionate (0.1 mg/μL, 20 μL/pellet)	♂ MV	↑ CoA, MeA 0 CeA, DG, HYP	--	0 AMY	[86]
	Estradiol benzoate (1.5 mg/pellet)	♂ MV	↑ CoA, MeA 0 CeA, 0 DG, HYP	early diff: 0 DG axon extension: ↑ DG maturation: 0 DG	0 AMY	[86,87]
	DHT (0.1 mg/μL, 20 μL/pellet)	♂ MV	0 AMY, DG, HYP	--	0 AMY	[86]
Long-term replacement (>14 days)	Testosterone proprionate (30-day:0.25 mg/0.1 mL or 100 mg/pellet)	♂ rat	0 GCL	0 GCL	0 GCL	[81,88]
	Testosterone proprionate (29-day: 0.5 or 1.0 mg/0.1 mL)	♂ rat	--	↑ GCL	0 GCL	[81]
	Testosterone proprionate (15- or 21-day: 0.5 or 1 mg/0.1 mL)	♂ rat	0 GCL	0 GCL	--	[83,89]
	17β-estradiol (0.01 or 0.02 mg/0.1 mL) Estradiol benzoate (15 μg/inj.)	♂ rat	0 GCL	0 GCL	0 GCL	[81,90]
	DHT (30-day: 0.25 and 0.5 mg/0.1 mL)	♂ rat (young & middle-aged)	--	↑ GCL	0 GCL	[81,91]

Abbreviations used: AMY, amygdala; AOB, accessory olfactory bulb; DG, dentate gyrus; CeA, central AMY; CoA, cortical AMY; diff, differentiation; DHT, dihydrotestosterone; GCL, granular cell layer in the dentate gyrus; HYP, hypothalamus; MeA, medial AMY; MeP, posterior medial AMY; MOB, main olfactory bulb; MPOA, medial preoptic area; MV, meadow vole; SGZ, subgranular zone in the dentate gyrus; SVZ, subventricular zone; ↑: increase; ↓: decrease; 0: no change; ↔: mixed findings; --: no data; ♂: male.

**Table 2 biomolecules-10-01151-t002:** Effects of estrogens on the different stages of adult neurogenesis in several mammalian species.

		Species	Cell Proliferation	Cell Survival	Neuronal Differentiation	References
**NATURAL INCREASE**						
Proestrus (highest level)		♀ rat	↑ DG 0 SVZ	↑ DG	0 DG	[109,111]
		♀ mouse	0 DG, 0 SGZ	--	--	[85,110]
		wild-living ♀ MV	↓ GCL, GCL + SGZ	--	↓ GCL + SGZ	[69,70]
		lab-reared ♀ MV	↓ DG	↓ DG	--	[112]
		♀ PV	↑ rms ↔ SVZ 0 AMY, DG	↑ AMY, HYP 0 DG, MOB	0 DG, MOB	[17,113]
**DECREASE VIA OVX**						
Short-term OVX (7 days)		♀ rat	↓ DG	--	--	[109,114]
Long-term OVX (>14 days)		♀ rat	0 DG	--	--	[111,114,115]
		♀ mouse	↔ DG	--	↔ DG	[85,110]
**OVX AND ESTROGEN REPLACEMENT**						
Acute replacement (7 days after OVX)	17β-estradiol (10 μg) or estradiol benzoate (10 μg) (30 min-4 hr prior to BrdU)	♀ MV, rat	↑ GCL, GCL + SGZ	--	--	[109,111,116,117,118,119,120,121]
	17β-estradiol (10 μg) or estradiol benzoate (10 μg) (48 hr prior to BrdU)	♀ rat	↓ DG	--	--	[116,117]
	17β-estradiol (0.3, 1, or 50 μg)	♀ rat	0 GCL	--	--	[111,120]
	17β-estradiol (1 or 10 μg, or pellet)	♀ mouse	↓ SVZ	↓ MOB	--	[122]
Acute replacement (>2 weeks after OVX)	17β-estradiol (10 μg)	♀ rat	0 DG	--	--	[111]
	Estradiol (100 μg/100 g)	♀ rat	↓ AOB 0 MOB	--	--	[123]
	Estradiol benzoate (1 μg/day 3 days)	♀ PV	↑ SVZ 0 rms, MOB	--	--	[113]
	Estradiol benzoate (pellet 48 h)	♀ PV	0 AMY, DG	--	--	[124]
	Estradiol benzoate (pellet 48 h)	♀ MV	↑ AMY 0 DG	--	--	[124]
Long-term replacement (>14 days)	Estradiol benzoate, estrone	♀ rat	0 GCL	↓ GCL	--	[87,125,126]
	17β-estradiol	♀ rat	0 GCL	↑ GCL	--	[125]
	17β-estradiol	♀ mouse	↑ARC, DMH, VMH	--	--	[127]

Abbreviations used: AMY, amygdala; ARC, arcuate nucleus of the hypothalamus; DG, dentate gyrus; GCL, granular cell layer in the dentate gyrus; DMH, dorsal medial hypothalamus; HYP, hypothalamus; MOB, main olfactory bulb; MV, meadow vole; PV, prairie vole; rms, rostral migratory stream; SGZ, subgranular zone in the dentate gyrus; SVZ, subventricular zone of the lateral ventricles; VMH, ventromedial hypothalamus; ↑: increase; ↓: decrease; 0: no change; ↔: mixed findings; --: no data; ♀: female.

**Table 3 biomolecules-10-01151-t003:** The effects of stressors on cell proliferation in several mammalian species.

	Stressor	Species	Impact	References
**ACUTE** (same day)				
Laboratory stressors	cold swim, foot shock, restraint, tail nick, tail shock	BALB/c mouse, Sprague-Dawley, wild house mouse, Wistar rat	↓ DG	[168,169,170,171,172,173,178,179]
Ethologically-relevant stressors	predator odor, social defeat	common marmoset (*Callithrix jacchus*), Sprague-Dawley rat, tree shrew (*Tupaia belangeri)*	↓ DG	[4,164,175,176,177]
Brain region-specific	Foot shock + restraint	BALB/c	↓ DG 0 SVZ	[168]
Sex-specific	foot shock, predator odor	Sprague-Dawley rat	♂: ↓ DG ♀: 0 DG	[171,176]
Species-specific	restraint	C57BL/6J mouse, Sprague-Dawley rat	mouse: ↑ DG rat: ↓ DG	[180]
Age-specific	footshock + restraint	C57BL/6N	adult: 0 DG aged: ↓ DG	[181]
**SUBCHRONIC** (<14 days)				
Laboratory stressors	7-day daily restraint	Sprague-Dawley rat	↓ DG	[182]
Ethologically-relevant stressors	5-, 7-, or 10-day daily social defeat	CFW mouse, C57BL mouse, Wistar rat	↓ DG	[183,184,185]
Brain region-specific	social defeat	C57BL mouse	0 AMY ↓ DG	[184]
**CHRONIC** (≥14 days)				
Laboratory stressors	14-day intermittent restraint stress; 20-day intermittent mild stress; 21-day daily foot shock; 14-, 21-, 42-, 49-, or 56-day of unpredictable stress; 21- or 42-day daily restraint stress	BALB/c mouse, Sprague-Dawley rat, Wistar rat	↓ DG	[122,170,178,186,187,188,189,190]
Ethological-relevant stressors	35-day dominance hierarchy; 14-day social defeat; 18- or 35-day of daily social defeat; 42-day social isolation	CD-1 mouse, C57BL/6J mouse, PV, tree shrew (*Tupaia belangeri),* Wistar rat	↓ DG	[191,192,193,194,195,196]
Brain region-specific	21-day unpredictable stress, 42-day social isolation, 49-day chronic mild stress	BALB/c, PV, Sprague-Dawley rat	↓ DG, MPOA 0 AMY, hilus, SVZ, VMH	[82,165,195]

Abbreviations used: AMY, amygdala; DG, dentate gyrus; MPOA, medial preoptic area; PV, prairie vole: SVZ, subventricular zone; VMH, ventromedial hypothalamus; ↑: increase; ↓: decrease; 0: no change; ↔: mixed findings; --: no data; ♂: male; ♀: female.

**Table 4 biomolecules-10-01151-t004:** The effects of stressors on cell survival in several mammalian species.

	Stressor	Species	Impact	References
**ACUTE** (same day)				
Laboratory stressor	tail shock	Sprague–Dawley rat	↓ DG	[174]
Ethologically-relevant stressors	predator odor, social defeat	Sprague–Dawley rat	↓ DG	[164,200]
**SUBCHRONIC and CHRONIC**				
Laboratory stressors	21-day daily restraint, 21-day chronic mild stress, 49-day chronic mild stress	BALB/c, CD 1 mouse, Sprague–Dawley rat	↓ DG	[80,82,158,165,178]
Ethologically-relevant stressors	10-, 18-, or 35-day daily social defeat, 42-day social isolation	C57BL/6, PV, Sprague–Dawley rat, Wistar rat	↓ DG	[87,192,195,205]
Brain region-specific	chronic mild stress, restraint	CD-1 mouse, PV, Sprague–Dawley rat	↓ AMY, DG, VMH 0 CA 1, CA 3, hilus, MPOA	[80,82,195]

Abbreviations used: AMY, amygdala; CA 1, CA region 1 in the hippocampus; CA 3, CA region 3 in the hippocampus; DG, dentate gyrus; MPOA, medial preoptic area; VMH, ventromedial hypothalamus: ↑: increase; ↓: decrease; 0: no change; ↔: mixed findings; --: no data.

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
