# Peer review of "Hormonal Regulation of Mammalian Adult Neurogenesis: A Multifaceted Mechanism"

_biomolecules, 2020, doi:10.3390/biom10081151_

Round 1
Reviewer 1 Report
Review of Jorgensen & Wang for Biomolecules
This review article is very well written and carefully organized. The authors identify a previously unreviewed niche and appropriately point this out in the introduction. The niche assesses the role of hormones on neurogenesis. The authors separate the discussions by the three stages of neurogenesis: cell proliferation, cell survival, and neuronal differentiation. In addition, neurogenesis is examined for its regulation by gonadal hormones, glucocorticoids, and other neurochemicals. The review systematically compares and contrasts many studies in an eloquent way. They conclude that hormones have a multifaceted impact on regulating neurogenesis. I only have a few major comments and several minor comments itemized below in chronological order.
Comments:
I would suggest adding a table of contents to aid the reader in grasping the scope of the article from the start.
Figure 1 uses abbreviations that are not defined until pages later or not at all. Abbr. not defined at all are GICL, MCL, GrCL. Abbr. defined further after the figure appears are AOB. I suggest defining abbreviations at an obvious location.
Figure 1, I suggest rearranging legend so that the AOB and MOB show up in the same order from top to bottom as they do on the figure. In other words, put MOB at the top.
On page 10 the statement, “-a concern that was not validated in a recent study” and the sentences around it are confusing. Why mention this alternative explanation about captivity if you are only going to present evidence that contradicts that very alternative explanation?
Also on page 10 the next paragraph starting with “using prairie voles…”, the authors mention the type of control group used as a potential problem. Then types of mesh barriers are discussed, which I assume pertains to the control groups in the studies cited. Am I assuming correctly? Could you be more detailed in your description of how these variables could explain the conflicting results?
Also on page 10, is the phrase “unrestricted interactions” referring to social interactions?
Also on page 10, the authors bring up prairie voles for the first time without any introduction to the difference between the meadow and prairie voles. There is one brief mention of a difference 8 paragraphs later, but I suggest placing this information before the different types of voles are mentioned.
At the top of table 3 on page 13 text should probably read “tail nick” instead of “…restraint, tail, nick, tail…”. Also, why capitalize footshock in the third row but nowhere else?
Table 3, I believe there are some formatting mistakes because the ACUTE subheading is centered and the other subheading Subchronic and Chronic are aligned left.
On the first line of Page 15, fix the grammar in “It is worth to mention that…”
I am assuming this is supposed to be an exhaustive literature review given the 298 references, but I believe the review could be more thorough. For example, why did you omit this relevant paper? C.R. McKittrick, A.M. Magariños, D.C. Blanchard, R.J. Blanchard, B.S. McEwen, R.R. Sakai, Chronic social stress reduces dendritic arbors in CA3 of hippocampus and decreases binding to serotonin transporter sites, Synapse 36 (2000) 85–94, https://doi.org/10.1002/(SICI)1098-2396(200005)36:2<85:AID-SYN1>3.0. CO;2-Y.
Capitalize heading for section 4.
In section 4.1.2, which has page number 3 at the bottom but is not actually the third page, why is 5-HT1A and 5-HT2C shown with the 1A and 2C as a subscript, but the 5-HT2A has no subscript?
In section 4.1.3, labeled page 4, I could not locate where OT is defined. I assume this is about oxytocin but no definition could be located.
I believe that a summary figure would be helpful in summarizing the main points.
In the acknowledgements section a supplemental figure, table and video are mentioned, but I could not review these. The website link does not direct me to these documents. I am going to assume this is a mistake since there was no mention of these supplements in the text of the review.
Author Response
- I would suggest adding a table of contents to aid the reader in grasping the scope of the article from the start.
Response: Thank you for the very helpful suggestion, such a table of content has been added (pages 3).
- Figure 1 uses abbreviations that are not defined until pages later or not at all. not defined at all are GICL, MCL, GrCL. Abbr. defined further after the figure appears are AOB. I suggest defining abbreviations at an obvious location.
Response: Thank you for noting this. We have included a list of abbreviations (pages 3-4) and included a more detailed figure caption (page 4). We hope this will make the long list of abbreviations more accessible.
- Figure 1, I suggest rearranging legend so that the AOB and MOB show up in the same order from top to bottom as they do on the figure. In other words, put MOB at the top.
Response: We have reversed the order, the legend for MOB appears now above the legend for the legend for AOB.
- On page 10 the statement, “-a concern that was not validated in a recent study” and the sentences around it are confusing. Why mention this alternative explanation about captivity if you are only going to present evidence that contradicts that very alternative explanation?
Response: Thank you for the suggestions. We have removed this alternative explanation.
- Also on page 10 the next paragraph starting with “using prairie voles…”, the authors mention the type of control group used as a potential problem. Then types of mesh barriers are discussed, which I assume pertains to the control groups in the studies cited. Am I assuming correctly? Could you be more detailed in your description of how these variables could explain the conflicting results?
Response: Thank you for pointing out this lack of clarity. We have added and/or changed wording to make this distinction clearer.
- Also on page 10, is the phrase “unrestricted interactions” referring to social interactions?
Response: ‘social’ has been added to improve clarity.
- Also on page 10, the authors bring up prairie voles for the first time without any introduction to the difference between the meadow and prairie voles. There is one brief mention of a difference 8 paragraphs later, but I suggest placing this information before the different types of voles are mentioned.
Response: Thank you for pointing out. We have added and/or changed wording (now on page 15) to add such an introduction to the two types of voles.
- At the top of table 3 on page 13 text should probably read “tail nick” instead of “…restraint, tail, nick, tail…”. Also, why capitalize footshock in the third row but nowhere else?
Response: Thank you for finding this error. It has been corrected.
- Table 3, I believe there are some formatting mistakes because the ACUTE subheading is centered and the other subheading Subchronic and Chronic are aligned left.
Response: This has been corrected.
- On the first line of Page 15, fix the grammar in “It is worth to mention that…”
Response: This has been corrected.
- I am assuming this is supposed to be an exhaustive literature review given the 298 references, but I believe the review could be more thorough. For example, why did you omit this relevant paper? R. McKittrick, A.M. Magariños, D.C. Blanchard, R.J. Blanchard, B.S. McEwen, R.R. Sakai, Chronic social stress reduces dendritic arbors in CA3 of hippocampus and decreases binding to serotonin transporter sites, Synapse 36 (2000) 85–94, https://doi.org/10.1002/(SICI)1098-2396(200005)36:2<85:AID-SYN1>3.0. CO;2-Y.
Response: Thank you for your wonderful suggestion. Our original draft was approximately 4000 words too long to fit the journal requirements. The current version is still on the lengthy side and I fear adding extra discussion might make the review too long.
- Capitalize heading for section 4.
Response: This has been corrected.
- In section 4.1.2, which has page number 3 at the bottom but is not actually the third page, why is 5-HT1A and 5-HT2C shown with the 1A and 2C as a subscript, but the 5-HT2A has no subscript?
Response: This has been corrected.
- In section 4.1.3, labeled page 4, I could not locate where OT is defined. I assume this is about oxytocin but no definition could be located.
Response: The abbreviation is mentioned in first paragraph of what is now section 5.1. on now page 24.
- I believe that a summary figure would be helpful in summarizing the main points.
Response: We have created and included a summary figure.
- In the acknowledgements section a supplemental figure, table and video are mentioned, but I could not review these. The website link does not direct me to these documents. I am going to assume this is a mistake since there was no mention of these supplements in the text of the review.
Response: Thank you for alerting us to this. You are correct. There are no supplemental figure, table, or video. We will try to make sure that this error will be corrected in the resubmission of the manuscript.

Reviewer 2 Report
In the manuscript entitled “Hormonal Regulation of Mammalian Adult Neurogenesis: A Multifaceted Mechanism” the authors reviewed on the impact of hormones on adult neurogenesis specifically the influence of gonadal hormones on adult neurogenesis, glucocorticoids on inhibition of neuronal differentiation, and neuroprotective effects of neurotrophins (BDNF) and neuromodulators (5-HT, Oxytocin etc). The review is well structured and authors literature is review is intense and insightful. However, I have minor concerns
- Authors mention Androgens facilitate DH neurogenesis (which is considered are abnormal neurogenesis) during peak breeding time. This give an impression that androgens facilitate abnormal neurogenesis.
- Effect of androgens during reproductive time, neural differentiation is less compared to non-breeding time. Authors should elaborate more on neurogenesis in non-breeding time.
- Effects of estrogens on the adult neurogenesis authors have speculative and contradictory statements on the facilitation of neurogenesis and neural differentiation by estrogens. Authors should clarify this for readers
- Since there are contradictory studies on the harmonal influence on cell proliferation, authors should provide more supporting evidence for their statements and should give their opinion in the end.
- Since studies show the impact of the hormones on the astrocytes and glial cells, authors should elaborate on the indirect effect of hormones on the adult neurogenesis, neural differentiation and cell proliferation via astrocytes and glia.
Author Response
- Authors mention Androgens facilitate DH neurogenesis (which is considered are abnormal neurogenesis) during peak breeding time. This give an impression that androgens facilitate abnormal neurogenesis.
Response: I am not sure I fully understand this comment. It is not clear what the abbreviation “DH” refers to. Furthermore, I am not sure what is meant by “abnormal neurogenesis.” Is this referring to abnormal levels, abnormal location, or maybe abnormal timing of adult neurogenesis?
- Effect of androgens during reproductive time, neural differentiation is less compared to non-breeding time. Authors should elaborate more on neurogenesis in non-breeding time.
Response: Thank you for your comment. We have tried to address your concern in two ways. (1) We have added wording to make it clear that the researchers only assessed correlation and not causation. (2) We also added wording to highlight that this correlational study was the only study to assess seasonal variations of androgen levels on neuronal differentiation.
- Effects of estrogens on the adult neurogenesis authors have speculative and contradictory statements on the facilitation of neurogenesis and neural differentiation by estrogens. Authors should clarify this for readers. Since there are contradictory studies on the harmonal influence on cell proliferation, authors should provide more supporting evidence for their statements and should give their opinion in the end.
Response: We agree with your observation that there are contradictory findings in regards to the hormonal influence on cell proliferation. We have polished wording in that section and included a figure (Figure 2) to improve the clarity of this section.
- Since studies show the impact of the hormones on the astrocytes and glial cells, authors should elaborate on the indirect effect of hormones on the adult neurogenesis, neural differentiation and cell proliferation via astrocytes and glia.
